# Evaluation of an enhanced service for medication review with follow up in Swiss community pharmacies: Pre-post study protocol

Noelia Amador-Fernández[1,2,3,4]*, Mathilde Escaith[1,2,3,4], Elodie Simi[1,2,3,4], Patricia Quintana-Bárcena[1,2,3,4,5], Jérôme Berger[1,2,3,4]

1 Centre for Primary Care and Public Health (Unisanté), University of Lausanne, Lausanne, Switzerland, 2 Centre for Research and Innovation in Clinical Pharmaceutical Sciences, Lausanne University Hospital and University of Lausanne, Lausanne, Switzerland, 3 Institute of Pharmaceutical Sciences of Western Switzerland, University of Geneva, University of Lausanne, Geneva, Switzerland, 4 School of Pharmaceutical Sciences, University of Geneva, Geneva, Switzerland, 5 Research Institute of the McGill University Health Centre, Montreal, QC, Canada

* noelia.amador-fernandez@unisante.ch

**Data Availability Statement:** No datasets were generated or analysed during the current study. All relevant data from this study will be made available upon study completion.

## Abstract

### Background

In Switzerland, 20,000 people are hospitalized each year as result of drug related problems (DRPs). The sources of DRPs can be related to patients' behavior (i.e., wrong administration) or to health processes (i.e., drug-drug interaction). No community pharmacy (CP) service focus on DRPs related to patients' behavior is currently recognized or remunerated in Switzerland. A medication review with follow up (MRF) has been developed to evaluate prescription and non-prescription medication.

### Objective

To evaluate the impact of MRF service for the identification and management DRPs associated to patients' behavior and to describe pharmaceutical interventions carried out through MRF.

### Methods

A pre-post intervention study with a cluster design and one intervention group will be carried out in CPs in the canton of Vaud (Switzerland) for 15 months. Volunteer pharmacists will be trained on the identification and management of DRPs related to patients' behavior. After training, they will include randomly selected adults taking four or more chronic drugs prescribed for at least three months prior to recruitment. Then, they will conduct three pharmacist-patient face-to-face consultations at 6-month intervals. Tasks will be differentiated by pharmacy technician or pharmacist to triage expired medication or to manage DRPs in a structured manner, respectively. The primary outcome is the identification of DRPs associated to patients' behavior. Secondary outcomes are to assess patients' medication

**Funding:** The study is funded by the local health authorities (DGS - Direction Générale de la Santé, www.vd.ch) and the pharmacist association (SVPh - Société Vaudoise de Pharmacie, www.svph.ch). The funders had not and will not have a role in study design, data collection and analysis, decision to publish, or preparation of the manuscript.

**Competing interests:** The authors have declared that no competing interests exist.

knowledge, number of expired medications, interventions carried out by pharmacists and pharmacists' satisfaction. The study will begin in April 2023 in 19 to 35 pharmacies that will recruit at least 162 patients. A sub analysis will be carried out for patients with 65 years old or over.

## Conclusions

The MRF intervention features a training designed for an enhanced evaluation of patient's behavior towards their medication. The study will allow the assessment and management of DRPs in Swiss CPs with the support of the local health authorities and pharmacist association.

**Trial registration:** Clinicaltrials.gov NCT05348538.

## Introduction

The early 2000s saw the emergence of new roles for community pharmacists in Switzerland, primarily in order to control rising health care costs [1, 2]. With the implementation of the "fee for service" in 2001, a step forward was taken in the direction of pharmacy services. Since then, it has been amended several times in order to respond to the health care system and patients' needs [3]. According to the Swiss Federal Council, pharmacists could still take on additional tasks in the field of primary care, as they offer easy access to health advice [4]. Since 2016, the canton of Vaud (French speaking part of Switzerland) has been supporting the implementation of a shared electronic medication plan concerning medication safety throughout the care process [5]. In this context, new opportunities exist to strengthen the role of the pharmacist in the process of care by sharing information related to the medication plan.

The clinical management of patients', especially polymedicated patients, goes through the identification and management of drug related problems (DRPs). The Pharmaceutical Care Network Europe (PCNE) defines a DRP as "an event or circumstance involving drug therapy that actually or potentially interferes with the expected optimal outcome of medical care" [6]. DRPs have several causes, whether related to the drug and dose selection, drug use process, patient or transfer related, etc. [6]. As an example, it has been estimated that the costs of DRPs such as non-adherence is approximately 125 billion euro per year in the European Union [7]. In Switzerland, each year, it is estimated that 20,000 hospitalizations result from DRPs [4]. Identification and management of DRPs related to health process such as drug-drug interactions, contraindications or abuse control are already remunerated as part of the "fee for services" related to prescription validation in Swiss community pharmacies (CPs) [8]. Nevertheless, DRPs related to the patient and his behavior, intentional or non-intentional, (i.e., patient stores drug inappropriately, administers/uses the drug in a wrong way, etc.) need a deeper understanding of the patient's specific situation and no service related to this aspect is currently available in Switzerland. For example, a treatment intake error could occur due to an instruction that is misunderstood / forgotten or due to improper medication storage at home. One way to identify and prevent the occurrence of DRPs in a structured manner is to conduct a medication review with the patient [9].

Medication review has been defined by the PCNE as "a structured evaluation of the patient's medications with the goal of optimizing their use and improving health status", which involves the detection of DRPs and the recommendation of interventions [10]. In addition, to increase

patients' safety, a medication review should include both, prescription and non-prescription medication, all of them reviewed by health professionals such as pharmacists for the identification of DRPs. Including non-prescription medication allows pharmacists to ensure responsible use of non-prescription medication [11, 12]. Indeed, community pharmacists are in a privileged position [13] to have an overview of all the medicines taken by patients (prescription and non-prescription) [14–16].

When prescribing, it is important that prescribers are aware of all medications, including non-prescription, in order to prevent interactions or to identify possible side effects: so, it is essential to support actions that allow pharmacists to share the information gathered through a medication review. It could be reach though the implementation of a shared electronic medication plan. Hence, it is important to develop a service that allows the inclusion of the most accurate information on all the medications a patient is taking on such a shared patient's medication plan.

The Cantonal Health Authorities of Vaud in collaboration with the Cantonal pharmacists association and the Center for Primary Care and Public Health of the University of Lausanne (Unisanté) developed an initiative for a medication review type 2A [17] with follow up (MRF) service. MRF includes follow up due to the results found in the literature, which suggests that medication review with follow up reduces patients' problems related with the use of medicines [18]. The aim of such service is to perform a medication review at the CP to 1) prevent, identify and manage DRPs related to patients due to the lack of such service in Switzerland, 2) improve patients' knowledge about their treatments and 3) remove of expired medications for proper elimination by the CPs. Based on the principles of the "brown bag" [19–21] and medication review with follow up [18, 22], MRF includes all the treatments consumed by the patient (prescription and non-prescription medication). This service allows the pharmacists going further than usual prescription validation in terms of detection of DRPs and pharmaceutical interventions. MRF offers the opportunity for the identification and management of DRPs related to patients, the assessment of all medication, disposal of expired medications and the detection of patients' lack of knowledge about their treatments. For this, the patient brings the medication to the CP before performing the service. MRF also strengthens the role of pharmacy technicians to perform an initial sorting of the medication brought to the CP to divide prescription and non-prescription and to eliminate expired medication. Therefore, allowing pharmacists an optimal use of the time devoted to the consultation. The service is in line with the national objective to reinforce pharmacy technicians' role, their education and to allow them to assist (under supervision) some services [23].

This paper describes a research protocol that aims to evaluate the impact of the MRF service for the identification and management of DRPs associated to patients' behavior in CP in Switzerland. This study also aims to describe pharmaceutical interventions carried out through MRF.

## Materials and methods

### Study design and setting

A pre-post intervention study with a cluster design and one intervention group will be carried out in Swiss CPs in the canton of Vaud. The study is taking place for 15 months between April 2023 and June 2024 and it will follow the standard protocol items SPIRIT [24] (see Fig 1 and S1 Appendix).

### Recruitment of study participants

Participant recruitment will take place at two different levels:

| | STUDY PERIOD | | | | | |
|---|---|---|---|---|---|---|
| | **Allocation** | **Before t0** | **Post-allocation** | | | **Close-out** |
| **TIMEPOINT** | *t-1* | **0** | *t0* | *t6* | *t12* | *t13* |
| **ENROLMENT:** | | | | | | |
| Eligibility screen | X | | | | | |
| Allocation | X | | | | | |
| Oral Informed consent | | X | | | | |
| **INTERVENTION:** | | | | | | |
| Medication review with follow up | | | 1st | 2nd | 3rd | |
| **ASSESSMENTS:** | | | | | | |
| Inclusion and exclusion criteria Participation | X | X | | | | |
| Service duration Patient related characteristics Patient medication list Patient's medication knowledge Removed medication Drug related problems Pharmacist's interventions | | | X | X | X | |
| Pharmacists' satisfaction | | | X | | | X |
| Pharmacy related characteristics Pharmacists related characteristics DRP severity | | | | | | X |

t0: first consultation with the patient
t6: second consultation with the patient, six months after the first one
t12: third consultation with the patient, twelve months after the first one

**Fig 1. Spirit schedule of enrollment, intervention and assessments [24].**

**CP and pharmacist level**: the Cantonal Health Authorities and the Cantonal pharmacists association will provide every CP in the canton of Vaud with written study information via email. In order to participate, CPs must have a consultation room and employ at least one pharmacist who will complete the training and will include at least one patient in the study. A CP may have different pharmacists who participate in the study; for those cases, all of them should complete the training and they have to choose a champion pharmacist [25] as a contact

person with the research team. A pharmacist who works in different participating pharmacies will be assigned to the CP where he/she works higher number of hours per week.

**Patient level**: patients will be recruited in the participant pharmacies (inclusion and exclusion criteria are listed below). To avoid selection bias by pharmacists selecting patients with more or less DRPs, pharmacists will list all patients who meet the inclusion criteria in each CP (S2 Appendix) and a member of the research team will pre-select 50 of them by using a sequence of computer-generated random numbers to be contacted by champion pharmacists. Pharmacists will follow the order in the list of the 50 randomly pre-selected patients to enroll from one up to ten. Flyers have been designed to help enrolling patients.

Eligible patients will be those adults taking four or more chronic drugs prescribed for at least three months prior to recruitment, irrespectively of the number of non-prescription medications. Exclusion criteria includes: patients suffering from dementia, psychiatric disorder, or other health condition that hinders obtaining informed consent and/or conducting the consultation with the pharmacist; patients who received a medication review within the last six months prior their study enrollment; patients who disagree meeting the pharmacist three times during the study with 6-month interval; patients who are not able to bring all their medication to the CP; patients who cannot speak and read French; patients who does not allow the pharmacist contacting the general medical practitioner (GP) to inform him/her about possible DRPs or patients who will not consent of participating in the study (patients who will not consent to participate will be able of accessing the service without being included in the study).

Eligible patients should meet inclusion criteria at the time of the first pharmacist-patient consultation. After the first consultation, those patients that have any treatment change and inclusion criteria is no longer met, can still be included in the study for the second and third consultation.

## Description of the intervention

The intervention is described using the TIDieR checklist [26] (S3 Appendix). MRF service will involve an intermediate medication review (type 2A as it is based on medication history and patient information) [17] with three pharmacist-patient face-to-face consultations in the CP consultation room at three stages of the study at 6-month intervals. Each of those three consultations will involve three stages: before, during and after the pharmacist-patient consultation. Different tasks performed by the pharmacist or pharmacy technician are also differentiated.

To ameliorate the delivery of the intervention and before the recruitment of patients, participating pharmacists will attend a half-day (four hours) training that will cover service provision, good practice standards, writing reports to patients and other health professionals, data collection and study protocol. The training will include a combination of lecture presentations and interactive sessions including role-play scenarios. The training will be accredited for pharmacists' continuing professional development and evaluated according to current standards. After the training, pharmacists in each participant CP will inform and train pharmacy technicians for their tasks using an online version of the training about the service and data collection (two online versions will be available, for pharmacists and pharmacy technicians). There will also have a web page that summarize the information [27]. In addition, participant pharmacists will be followed-up by the research team via telephone at least six times during the study (1st, 2nd, 3rd, 6th, 12th and 15th months of the study) to support them through the delivery of the service. Pharmacists will also be able to contact the research team by email throughout the study concerning service provision, data entry, etc. The facilitation process will be explained to pharmacists during the training session to ensure recruitment targets, quality of service provision and fidelity to study protocol are met.

**Before the pharmacist-patient consultation**: the purpose of this step is to prepare the necessary documents and the medication for the pharmacist-patient consultation. Patients will be asked to bring all their medication (prescribed and non-prescribed) to the CP. It will be noted if they have performed a pre-selection of the medication before bringing them to the CP. A pharmacy technician will pre-fill the documents with the patient information (medication plan, S4 Appendix). Pharmacy technicians will also review the medication brought by the patient to divide it in two groups (prescription and non-prescription medication) to identify the prescribed medication currently included in the medication plan and expired medications for the pharmacist-patient consultation. Then, pharmacists must have at least 30 minutes to review the prescription medication and non-prescription medication currently taken by the patient to assess potential DRPs before the consultation.

**During the pharmacist-patient consultation**: the purpose is to systematically identify potential DRPs related to patients' behavior and discuss all DRPs identified (before and during the consultation) to propose interventions with the patient or the GP to resolve them. Also, during the consultation, the pharmacist will evaluate patients' knowledge about their medication to improve it, if necessary.

**After the pharmacist-patient consultation**: within a maximum of two working days' period after the consultation with the patient, pharmacists will send a medication plan to both, patient and GP. Additional information will be sent to the GP in case his/her intervention is needed to solve some detected DRPs.

## Data collection methods

Pharmacists will collect data during their intervention at three time points during the study: at recruitment time, six months later and twelve months after the first encounter. A validated tool will be used for identifying and documenting DRPs, the PharmDISC tool [15] (PharmDISC tool will be used only including the 15 DRPs related to patients' behavior out of the 24 total DRP). In addition, a questionnaire including seven questions to evaluate patients' medication knowledge will be used. Pharmacist may choose to complete all documents (medication plan and patients' medication knowledge questionnaire which are merged in a single document, and, if any DRP is detected, the PharmDISC tool) in paper or electronic version during each consultation with the patient.

Participant pharmacists must complete a sociodemographic questionnaire about CP characteristics and his/her individual characteristics during the educational training. This questionnaire on paper will be collected at the end of the study. In addition, pharmacists should complete an electronic anonymous satisfaction questionnaire after each pharmacist-patient consultation. The time taken per patient to deliver the intervention (differentiated between pharmacist and pharmacy technician time) will be recorded. After coding data, champion pharmacists will send a copy of the data collected to the research team within one month after each consultation and/or contacting the GP when necessary. The research team will evaluate the documents to ensure quality of service provision and fidelity to the study protocol. Pharmacists will receive a remuneration of CHF 100 per pharmacist-patient consultation (which cannot be financed by the basic health insurance) and an additional amount of CHF 10 for the additional time spent on documentation of consultations and interventions for research purposes according to the guidelines given in the educational training. The amount of CHF 100 was calculated by the time expected for each professional involved in the service: approximately 45 minutes for pharmacists (CHF 87 per hour) and approximately 45 minutes for pharmacy technicians (CHF 53 per hour).

It is planned that a short communication will be done by the Cantonal Health Authorities; in addition, pharmacists will have flyers (with logo from the Cantonal Health Authorities) to

give to patients that will reinforce the message related to the intervention. There will be also a communication from the pharmacists' association that will go to the GPs association.

## Study measurements and outcomes

Dependent study outcomes and variables are presented in Table 1. In addition, independent variables will be collected in relation to patients (socio-demographic characteristics variables: gender, year of birth and education level and treatment variables: brand name/DCI, dose, formulation, posology for each medicament); pharmacists (socio-demographic characteristics: gender and year of birth and academic and work-related characteristics: role (manager/not manager), year of graduation in pharmacy, place of graduation in pharmacy, working hours in the CP) and CPs (average prescriptions per day, opening hours, location (urban/suburban/rural/industrial area), number of pharmacy technicians and pharmacists employed).

## Sample size

Primary objective of the study aims to measure the difference in DRPs per patient observed between consultations. Sample size calculation was based on this primary study outcome to detect a difference of 0.5 DRP per patient at the end of the study (using a one-tailed test because the hypothesis is that patients will have a decrease of DRPs at the end of the study). The calculation was based on a pilot study carried out on 20 patients at Unisanté CP. Sample size was calculated with $\geq 0.8$ power, type I error rate of 5%, assuming an intra-cluster correlation of 0.02. Allowing for 15% dropout, the overall sample size is 162 patients, with 19 to 35 pharmacies (5–10 patients per CP).

## Statistical methods and analysis

Continuous variables will be summarized using mean and standard deviation, or median and percentiles depending on the distribution of the variable. To assess normality, the Kolmogorov Smirnov test will be used. Categorical variables will be reported by frequency and proportion. For the comparison of continuous variables, the T Student's test or the ANOVA test will be carried out if there is a normal distribution, and Kruskal-Wallis otherwise. The comparison of the categorical variables will be carried out using the $\chi 2$ test, the Fisher's exact test or Yate's chi-squared test if necessary.

 For each patient, differences between consultations (second and first consultations, third and first consultations) will be compared for dependent variables (number of DRPs and medication expired, patient knowledge score, pharmaceutical interventions and medication removed, proportion of pharmaceutical interventions acceptance). A linear regression model will be performed accounting for the cluster effect. A cluster effect is expected by the way pharmacists interact with their team and the surrounding GPs, this could influence the number and type of DRPs detected and interventions made. A sub analysis will be carried out to evaluate the effect of patients undertaking or not a pre-selection of their medication before attending the pharmacy. Firstly, variables will be considered significant (p-value<0.2) in a bivariate model to be included in a multivariate model. Secondly, the variables considered significant at a value of p<0.1 in the multivariate model will be included in the final model. The level of significance will be set at p<0.05; the software STATA 17® will be used. An analysis with completed cases will be made (patients with data for the three consultations). In addition, an analysis with the whole sample will be made where, for the management of missing data, initial value will be considered as not been modified (baseline observation carried forward-BOCF) [28]. In case of a sufficient number of patients who are 65 years old or over are randomly

**Table 1.  Study outcomes.**

| Primary outcomes (type of variable) | Definition and assessment | Time point |
|---|---|---|
| Number of DRPs detected (quantitative) | DRPs associated with patients' behavior detected by the pharmacist before the consultation (evaluation of the medication) and during the consultation (evaluation of patients' information). Total number of DRPs and mean per patient and depending on the number of the patients' medication will be calculated. | Pharmacists will complete the medication plan and PharmDISC tool before and during the consultation. |
| Evolution on the number of DRPs (quantitative) | Evolution on the number of DRPs associated with patients' behavior detected will be assessed by comparing the differences between the three consultations with the patient. | Research team will calculate the differences using the medication plan at each consultation. |
| DRP classification (qualitative) | PharmDISC tool will be used for DRP related classification (15 DRPs related to patients' behavior out of the 24 total DRP included in the tool):<br>1. Drug choice: duplicate, accumulation, adverse event.<br>2. Dose choice: under dose/overdose.<br>3. Drug use: a. inappropriate timing or frequency of administration, b. inappropriate use method, c. inappropriate treatment duration.<br>4. Patient: insufficient adherence, insufficient knowledge, burden due to therapy, inappropriate stockage, absence of label.<br>5. Logistics: prescribed drug not available, error in medication process.<br>6. Other (to be detailed by the pharmacist). | Pharmacists will complete the PharmDISC tool before and during the consultation. |
| Number of pharmaceutical interventions (quantitative) | Number of interventions carried out by the pharmacist to correct DRPs and/or prevent their consequence.<br>Total number of interventions and mean per pharmacist, per patient, per number of medications and number of DRPs identified will be calculated. | Pharmacists will complete the PharmDISC tool before and during the consultation. |
| Pharmaceutical interventions classification (qualitative) | Classification of interventions will be made using PharmDISC tool:<br>• Substitution<br>• Dose adjustment<br>• Adjustment of the delivery amount (package size, quantity of packages, etc.)<br>• Optimization of administration / Change of administration way<br>• Therapy stopped/no delivery<br>• Therapy started<br>• Counselling of patient<br>• Application/use instruction (training)<br>• Delivery of an adherence aid inclusive counselling<br>• Clarification in the case notes (history)<br>• Information to the GP<br>• Therapy monitoring<br>• -Other | Pharmacists will complete the PharmDISC tool before and during the consultation. |
| Pharmaceutical interventions acceptance (quantitative) | Pharmaceutical interventions acceptance by the patient or the GP (when proposed to the GP, it will be considered as accepted unless the GP states otherwise). Number and proportion of intervention accepted versus proposed according to the number of medications and the number of DRPs. | Pharmacists will complete the PharmDISC tool during the second and third consultation. |
| **Secondary outcomes** | **Definition and assessment** | **Time point** |
| Number of removed medications (quantitative) | Expired or in bad condition or unused medications will be identified after reviewing the medication brought by the patient to the pharmacy: total number, mean per patient and percentage in relation to the total medication brought by the patient will be documented. | Pharmacists will complete the medication plan during the consultation with patients. |
| Number of removed medications discharged (quantitative) | Number of medications proposed to be removed and accepted by the patient after pharmacist-patient consultation: total number, mean per patient and percentage in relation to the total expired medication identified will be documented. | Pharmacists will complete the medication plan during the consultation. |

(*Continued*)

**Table 1.** (Continued)

| Primary outcomes (type of variable) | Definition and assessment | Time point |
|---|---|---|
| Patient's knowledge (quantitative) | Medication knowledge evaluation questionnaire with 7 questions (Why do you take this medication?; When do you take this medication?; How many tablets/pushes/milliliters do you take per dose?; How do you take this medicine with food?; What do you do if you forget to take a dose of this medicine?; What side effects may occur with this medicine?; At home, where do you keep this medicine?). The patient scores 1 point if he/she knows the correct answer, 0 points if he/she does not know and -1 if the information given is incorrect. Score goes from -7 (patient who answers all questions incorrectly) to 7 (patient who answer all questions correctly) for each medication that the patient is taking. A score of 0 has no particular significance, apart from being in the middle of the two extremes.<br>The questionnaire is being validated nowadays in a separate project. It aims to evaluate patient's knowledge over time, rather to differentiate between patients.<br>The total score and the mean score (total score divided by the number of medicines) will be documented. | Pharmacists will complete the questionnaire during the consultation (included in the medication plan). |
| Pharmacists' satisfaction (qualitative) | Pharmacists' satisfaction with MRF will be measured using a questionnaire submitted (8 questions using a Likert scale from "Totally disagree" to "Completely agree").<br>Once received by the research team, all questionnaires will be coded. | Pharmacists will complete after the first pharmacist-patient consultation and at the end of the study. |
| **Other outcomes** | **Definition and assessment** | **Time point** |
| Participation (quantitative) | In order to test the interest and feasibility of the service, participation rate will be calculated as the number of patients who accepted to participate in the study of those who were proposed to participate in the study after randomly pre-select patients who meet the inclusion criteria. The reasons for refusal will also be documented. | Pharmacists and pharmacy technicians will complete an inclusion document at the beginning of the study, listing all patients in the pharmacy following the inclusion criteria. |
| Service duration (quantitative) | Time (minutes) invested by the pharmacists and the pharmacy technicians to deliver the intervention (recorded separately). Time invested at each consultation and total duration will be considered. | Pharmacy technicians (before the consultation) and pharmacists (after the consultation) will complete the medication plan. |
| DRP severity (qualitative) | Severity categorization of pharmacists' evaluation following SCOPE criteria. Severity can be classified as mild (I, II), moderate (III, IV) and severe (V and VI). | Research team will categorize the severity of DRPs using the PharmDISC tool completed by the pharmacist at the consultation. |

included in the study, a sub analysis will be carried out to target the intervention to a specific group of patients with higher risk for DRPs [29, 30].

## Data management

Champion pharmacists are responsible for keeping all documents in their CP according to current legal practices and requirements. The different forms used during the study will be recorded in the pharmacy using an electronic format (Microsoft Excel®) and they will be coded. Study data will be collected and managed using Research Electronic Data Capture (REDCap 12.5.4© 2022 Vanderbilt University) hosted at Unisanté [31] to improve data privacy and data reliability. Data will be double checked by an external data processing agency to assure codification is being done correctly by pharmacists. Afterwards, data will be accessible by the research team at Unisanté via RedCap®. Data will only be accessible by the researchers involved in the project and will be hosted on Unisanté's server (backed up electronically). At the end of the study, data will be kept up to 10 years after the last publication.

## Status and timeline

Data collection started in April 2023. Statistical analysis will be completed in summer 2024. Following this, the findings will be disseminated through a final report and communications to the health professionals and general population. In addition, congress communications and peer-review publications in scientific journals. It is also a will of the local association and authorities to continue with the service after the study.

## Ethics approval and safety considerations

The study was approved by the Cantonal Commission for Ethics in Human Research (CER-VD) (registration number 2023–00139, protocol version 04, 21.03.2023) (S5 Appendix). Participants will give oral consent; a patient could withdrawal from the project due to removal of informed consent, change of the community pharmacy (for a pharmacy not involved in the project, i.e., out of the Canton), absence to pharmacist-patient consultations, or death. In case of withdrawal of informed consent, data will be destroyed (by identifying patient's Id), therefore patient's data will not be used in the analysis and study results.

## Trial registration

The protocol of the study was registered on ClinicalTrials database in May 2022 under the number NCT05348538. In addition, S6 Appendix includes the items from the World Health Organization Trial Registration Data Set.

## Discussion

The present protocol describes the design of a pre-post intervention study in Swiss CP to determine the impact of an enhanced service (Fig 2), which is defined as service that "transcend conventional requirements of an outpatient pharmacy program contract that are focused on improving clinical and global patient outcomes" [32], for MRF in order to detect DRPs. The study reports one primary outcome measure, the identification and management of DRPs related to patients with polypharmacy.

A service based on medication review was part of the remunerated services listed in Switzerland [33], which focused on adherence and patients' knowledge [9, 33], but the reimbursement of the service stopped in 2019. Messerli et al. evaluated the service in Switzerland and

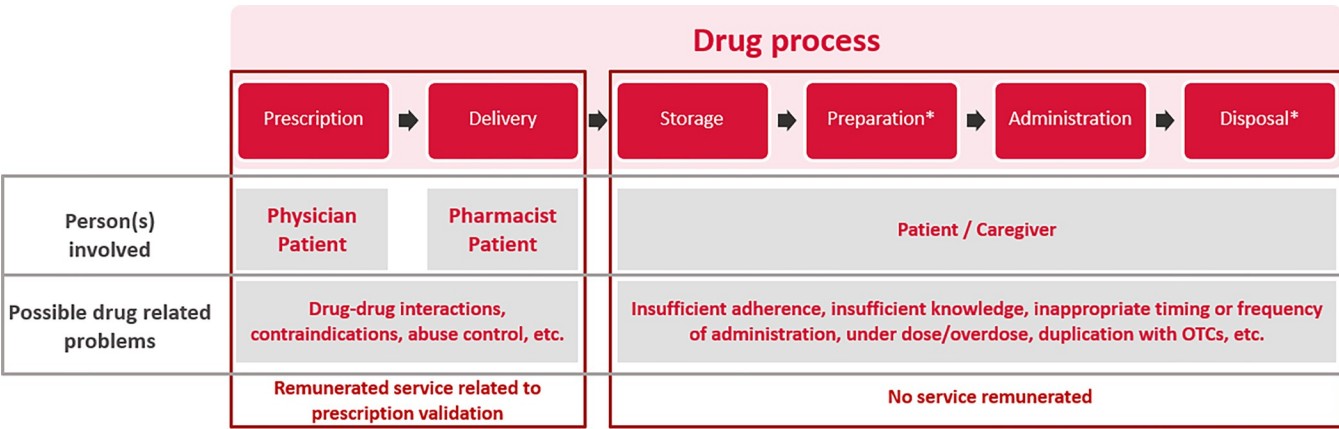

**Fig 2. Possible DRPs through the drug process according to remunerated Swiss community pharmacy services.**

made some recommendations for future services: include validated, structured and standardized interventions, tailor the service to higher risk patients for DRPs and include training and supervision for CP in order to improve service provision [9, 22]. General recommendations about the medication review service were also included in the National Institute for Health and Care Excellence (NICE) guidelines indicating that "all prescribed, over-the-counter and complementary medicines that the person is taking or using should be taken into account" [34]. In addition, WHO recommendations for health workers interventions includes the "use of individuals' home-based record to monitor their health status and identify potential health conditions" [35]. The study focusses on a service (MRF) designed to overcome those barriers found in the implementation process of medication review services. It includes an adapted tool, PharmDISC [15] for the evaluation and classification of DRPs related to patients' behavior and pharmaceutical interventions to deliver better care; it includes pharmacists' training at the beginning of the study and facilitation throughout; it also includes an evaluation of patients' self-medication. No amendments or modifications are planned to be made in the intervention during the course of the study. Therefore, the practice change facilitators and the research group will evaluate throughout the study if the service is provided as planned (fidelity of the intervention).

## Comparison with literature

Medication review with follow up reduces patients' problems related with the use of medicines [18]. Although it will not be included in the study, MRF has also suggested other positive impacts in test labs such as glycosylated hemoglobin, blood pressure or cholesterol [36], avoidance of medication-related hospitalizations [37, 38] and it has positive effects across clinical, economic, and humanistic outcomes [22].

A sub analysis for elderly people is proposed due to the increase in life expectancy resulting in a greatest prevalence for using a high number of medicaments for elderly people [18, 38]. For the evaluation of the present study, a pre-test/post-test design is used, considering CPs as cluster of the study to decrease the potential for contamination.

Some limitations to the study should be discussed. First, the study will be conducted in the canton of Vaud, thus, the results obtained may not be applicable to the rest of Switzerland. Indeed, national (federal) but also local (cantonal) laws and ordinances govern pharmacy practice in Switzerland. Thus, it must be applicable with some considerations especially for those regions where the GP prescribes and dispense medication (currently performed in some German speaking cantons) [39]. To minimize the impact on the generalization of the results obtained to the target population, as it have been noticed in the literature [40], patients included in the study will be randomly preselected by the research team (to avoid selection bias by pharmacists selecting only patients who are more likely to have better results). Second, the patient may not bring all the medications at the time of the consultation, which could lead to an underestimation of the identification of DRPs. However, pharmacy technicians will insist in the importance of reviewing all the medication with the patient. Third, dropouts are expected due to the number (three) of consultations with the patient and the duration of the study (fifteen months), although different analysis will be carried out (complete cases and all sample) to study possible differences and bias. Also, considering there will be a pre-post measurement of outcomes with no control group, it will not be possible to compare the MRF service directly to the current Swiss pharmacy practice. Nonetheless, outcomes will be measured at 6-month intervals, which will allow establishing if the impact of the MRF service varies across time. And lastly, as any other study, the Hawthorne effect may influence patients and pharmacists, that is, the effect of being studied which can potentially impact on participants' behavior.

## Conclusion

The MRF service responds to several needs to perform medication reviews in Swiss CPs. The MRF intervention features a training designed for the enhanced evaluation by both community pharmacists and pharmacy technicians of patient's behavior towards medication. This study will allow the assessment for the prevalence of DRPs and the feasibility of implementing the MRF service in Swiss CP with the support of the local health authorities and pharmacist association.

## Supporting information

**S1 Appendix. Recommended items to address in a clinical trial protocol and related documents.**
(PDF)

**S2 Appendix. Form used by pharmacists to included all patients in the pharmacy who meet the inclusion criteria.**
(PDF)

**S3 Appendix. Template for intervention description and replication checklist.**
(PDF)

**S4 Appendix. Form used by pharmacists and pharmacy technicians to list and evaluate patients' medication (original version in French).**
(PDF)

**S5 Appendix. Research protocol accepted by the ethics commitee.**
(PDF)

**S6 Appendix. Items from the World Health Organization trial registration data set.**
(PDF)

## Author Contributions

**Conceptualization:** Noelia Amador-Fernández, Mathilde Escaith, Elodie Simi, Patricia Quintana-Bárcena, Jérôme Berger.

**Funding acquisition:** Jérôme Berger.

**Methodology:** Noelia Amador-Fernández, Mathilde Escaith, Elodie Simi, Patricia Quintana-Bárcena, Jérôme Berger.

**Project administration:** Noelia Amador-Fernández, Mathilde Escaith, Elodie Simi, Patricia Quintana-Bárcena, Jérôme Berger.

**Software:** Mathilde Escaith.

**Supervision:** Noelia Amador-Fernández, Patricia Quintana-Bárcena, Jérôme Berger.

**Validation:** Mathilde Escaith.

**Writing – original draft:** Noelia Amador-Fernández, Jérôme Berger.

**Writing – review & editing:** Noelia Amador-Fernández, Mathilde Escaith, Elodie Simi, Patricia Quintana-Bárcena, Jérôme Berger.

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
