## [Decision Letter · Decision Letter 0]

11 Jul 2023

PONE-D-23-10141Evaluation of an enhanced service for medication review with follow up in Swiss community pharmacies: pre-post study protocolPLOS ONE

Dear Dr. Amador-Fernández,

Thank you for submitting your manuscript to PLOS ONE. After careful consideration, we feel that it has merit but does not fully meet PLOS ONE’s publication criteria as it currently stands. Therefore, we invite you to submit a revised version of the manuscript that addresses the points raised during the review process.

The protocol has been reviewed by the peer reviewers, one with a focus on research methods and one with a more clinical focus. Please make sure that all remarks are considered in the possible revision.

We look forward to receiving your revised manuscript.

Kind regards,

Sascha Köpke

Academic Editor

PLOS ONE

2. We note that the original protocol that you have uploaded as a Supporting Information file contains an institutional logo. As this logo is likely copyrighted, we ask that you please remove it from this file and upload an updated version upon resubmission.

Reviewers' comments:

Reviewer's Responses to Questions

**Comments to the Author**

1. Does the manuscript provide a valid rationale for the proposed study, with clearly identified and justified research questions?

Reviewer #1: Partly

Reviewer #2: Yes

2. Is the protocol technically sound and planned in a manner that will lead to a meaningful outcome and allow testing the stated hypotheses?

Reviewer #1: Partly

Reviewer #2: Yes

3. Is the methodology feasible and described in sufficient detail to allow the work to be replicable?

Reviewer #1: No

Reviewer #2: Yes

4. Have the authors described where all data underlying the findings will be made available when the study is complete?

Reviewer #1: Yes

Reviewer #2: Yes

5. Is the manuscript presented in an intelligible fashion and written in standard English?

Reviewer #1: Yes

Reviewer #2: Yes

6. Review Comments to the Author

You may also provide optional suggestions and comments to authors that they might find helpful in planning their study.

Reviewer #1: This is an interesting study looking reviewing of medication in order to identify and management drug related problems associated to community pharmacy and commend the research team.

Some comments:

The design has been stated to be a pre-post intervention study, which is a cluster design with community pharmacist as the unit, followed by patients recruited to each CP.

1. Suggest to include "cluster design" under the design and setting section.

2. Who generated the random numbers, e.g an independent team member not involved in recruitment

3. Table 1 - Study outcomes, make primary, secondary outcomes distinct as table because with a feasibilty assessment/

4. The sample size calculation states that the study was designed to detect 0.5 DRP per patient. Explicitly include what 0.5 DRP means exactly, i.e DRPs based on PharmDISC tool between consultations, or the PharmDISC tool - and only using 15 DRPs related to patients behavior out of 24. Or an literal number of DRPs - the definition needs more clarity in the manuscript.

5. Considering the selection of participants is random, how would you handle representation of the sample population, say in terms of important social demographic factors, unless is there any evidence to suggest that they are no differential rates of DRPs in certain groups of individuals? Have the researchers considered a possible bias in the selected individuals for this study? This becomes more apparent on this point, as part of the analysis will involve looking at a subgroup of patients

6. 1 or 2-sided significant testing - include in sample size section

Reviewer #2: It is a very interesting protocol. My comments are:

LINE 60; The definition of MR by PCNE includes different types of MR. It should be interesting to explain which one is the authors using in this project.

LINE 76: After explaining the use and impact of MR service, the authors add up a follow-up. At the end of the paper, under the section Comparison with literature, they explain why adding follow up to MR is useful. In my opinion here, at the beginning of the paper, the explanation fit better.

LINE 84: I would add an “of”. …, the assessment OF all medication….

LINES 92-95. The objective of the protocol is to evaluate DRPs associated to patients´ behavior. However, the authors haven explained which type o DRPs are those. During the protocol, when talking about PharmDISC, they explain that the tool includes 24 DRPs, but only 15 are related to patients´ behavior. Again, I think this explanation should be included in the introduction.

LINE 93: I would add an “of”. …, identification and management OF DRPs…

LINE 99: Work schedule should be reviewed and readjusted. (Currently we are on July, 2023)

LINE 111: I would change the comma to a semicolon. (…who participate in the study; for those cases, all…)

LINE 121: When talking about eligible patients, It is not clear to me the role of non-prescription drugs if only patients with 4 chronic prescription drugs (or more) can enter the study.

LINE 130: I would add a “d” at the end of “include”. … the service without being included in the study.

LINE 169. Again, the authors talk about DRPs related to patients` behavior. It should be explained in the introduction section.

LINE 180: The authors reference the PharmDISC tool using the Reference#22, but in LINE 298 they use the reference #15 to do the same.

LINE 202: I would change the comma to a semicolon. (Cantonal Health Authority; in addition…)

TABLE 1: Patients´ knowledge. The authors state that the Medication Knowledge Questionnaire is being validated, which is OK. However, I would like to know a bit more about the scoring system from -7 to 7. in this context, what would a score of zero mean?

In this same box, the phrase “How do you take this medicine with food” is duplicated and should be deleted.

TABLE 1: Number of Pharmaceutical. I wonder if the term “Number of Pharmacist Interventions” should fit better than “Number of pharmaceuticals”, but as this is an English wording question, I let it to the authors to discuss about.

TABLE 1: Pharmaceutical interventions acceptance. I would add an “s”. ..proportion of interventionS accepted versus…

LINE 244: I would change the comma to a semicolon. …p<0.05; the software STATA…

LINE 261-263: As said previously, work schedule should be reviewed and readjusted, as we currently are on the month of July.

LINE 270: I would change the comma to a semicolon. … oral consent; a patient could withdrawal…

LINES 292-293: I would change a little bit the wording: … the National Institute for Health and Care Excellence (NICE) guidelines indicating that “all…

LINE 301: I would change the comma to a semicolon. And facilitation thorough; it also includes…

7. PLOS authors have the option to publish the peer review history of their article (what does this mean?). If published, this will include your full peer review and any attached files.

Reviewer #1: No

Reviewer #2: **Yes: **miguel angel gastelurrutia

---

## [Author Response · Author response to Decision Letter 0]

24 Jul 2023

Thank you very much for the time and thorough review, we have answered each one of the comments separately (see also table attached):

Reviewer 1

1. The design has been stated to be a pre-post intervention study, which is a cluster design with community pharmacist as the unit, followed by patients recruited to each CP. Suggest to include "cluster design" under the design and setting section. 

Response to reviewers: Following the reviewer’s comment, a clarification was included about the study design. 

Modifications in the manuscript: 

A sentence was modified in the abstract (line 14) and the study design and setting’s section (line 103): A pre-post intervention study with a cluster design and one intervention group will be carried out […]

2. Who generated the random numbers, e.g an independent team member not involved in recruitment 

Response to reviewers: Pharmacists must send to the research team a coded list of all patients in the community pharmacy with the inclusion criteria (it is a list generated by the pharmacy software according to patients’ age and number of prescribed chronic medications).

From each community pharmacy list, a member of the research team will pre-select fifty patients by using computer-generated random numbers.

Then the pharmacists will call patients following the order included in the list with the 50 pre-selected patients. Pharmacists will include up to 10 patients. If a patient refuses to be included, the reason will be documented. 

Modifications in the manuscript:

Sentences were modified in the recruitment of study participants’ section (lines 123-127): Pharmacists will list all patients who meet the inclusion criteria in each CP (S2 Appendix) and a member of the research team will pre-select 50 of them by using a sequence of computer-generated random numbers to be contacted by champion pharmacists. Pharmacists will follow the order in the list of the 50 randomly pre-selected patients to enroll from one up to ten.

3. Table 1 - Study outcomes, make primary, secondary outcomes distinct as table because with a feasibilty assessment/ 

Response to reviewers: Following the reviewer’s comment, the order of variables in table 1 was changed to differentiate between primary, secondary and other outcomes. 

Modifications in the manuscript: Table 1 was modified.

4. The sample size calculation states that the study was designed to detect 0.5 DRP per patient. Explicitly include what 0.5 DRP means exactly, i.e DRPs based on PharmDISC tool between consultations, or the PharmDISC tool - and only using 15 DRPs related to patients behavior out of 24. Or an literal number of DRPs - the definition needs more clarity in the manuscript. 

Response to reviewers: One of the primary outcomes of the study is the evolution on the number of DRPs, therefore the sample size was calculated considering that the number of DRPs will decrease in 0.5 between the first and last consultation with the pharmacist.

The reason for using only 15 DRPs of the 24 DRPs included in the PharmDISC is due to the aim of the service, which aims to detect DRPs related only to patients’ behavior. 

The rest 9 DRPs are already evaluated in community pharmacy in Switzerland. As included in the introduction section (lines 52-54):

“DRPs related to health process such as drug-drug interactions, contraindications or abuse control are already remunerated as part of the fee-for-services related to prescription validation in Swiss community pharmacies”.

A figure was included in the introduction section to show the DRPs based on the drug process and the services remunerated in Switzerland (see attached). 

Modifications in the manuscript: A figure (Figure 1) was included in the introduction section (see attached).

5. Considering the selection of participants is random, how would you handle representation of the sample population, say in terms of important social demographic factors, unless is there any evidence to suggest that they are no differential rates of DRPs in certain groups of individuals? Have the researchers considered a possible bias in the selected individuals for this study? This becomes more apparent on this point, as part of the analysis will involve looking at a subgroup of patients 

Response to reviewers: The service aims to evaluate patients with polypharmacy, that is reason why any patient with a minimum of four prescribed medications can be included. 

Multivariate analysis will be carried out considering all patients’ characteristics to allow comparison in the number and evolution of DRPs between individuals.

Evidence suggests that patients who are 65 years old or over have higher risk of having DRPs. In case, there is enough number of patients who are 65 years or older randomly included in the study, the sub analysis will be carried out. A sentence was modified in the methods section.

The most important selection bias (pharmacists selecting patients with more or less DRPs) was considered in the protocol. So in order to avoid it , we decided to randomized patients. Pharmacist identified ALL patients with inclusion criteria in their pharmacies which represents all patients eligible for the service, and then the research team randomized them to include a representative sample in the study. It was considered the way of including a sample that represents those patients over 18 years old with four or more chronic medications who attend community pharmacy. A sentence was included in the methods section. 

Modifications in the manuscript:

A sentence was modified in the recruitment of study participants’ section (lines 122-123): To avoid selection bias by pharmacists selecting patients with more or less DRPs, pharmacists will list all patients […]

A sentence was modified in the statistical methods and analysis’ section (lines 260-262): In case of a sufficient number of patients who are 65 years old or over are randomly included in the study, a sub analysis will be carried out to target the intervention […]

6. 1 or 2-sided significant testing - include in sample size section 

Response to reviewers: The hypothesis is that patients will have a decrease of the number of DRPs at the end of the study, after the three consultations with the pharmacist, therefore, one sided test was used. A sentence was included following the reviewer’s comment. 

Modifications in the manuscript:

A sentence was modified in the sample size’s section (lines 232-233): (using a one-tailed test because the hypothesis is that patients will have a decrease of DRPs at the end of the study).

Reviewer 2

LINE 60; The definition of MR by PCNE includes different types of MR. It should be interesting to explain which one the authors is using in this project. 

Response to reviewers: Following the reviewer’s comment, the type of MR was included in the manuscript. 

A type 2A or intermediate MR is performed as it is based on the medication history and the patient’s information to evaluate for example adherence issues or problems with OTC. 

Modifications in the manuscript:

A comment was added in the introduction (line 78): […] developed an initiative for a medication review type 2A (16) with follow up (MRF) […]. 

A sentence was added in the description of the intervention (lines 146-147): MRF service will involve an intermediate medication review (type 2A as it is based on medication history and patient information) (16) with three pharmacist-patient face-to-face consultations […]. 

LINE 76: After explaining the use and impact of MR service, the authors add up a follow-up. At the end of the paper, under the section Comparison with literature, they explain why adding follow up to MR is useful. In my opinion here, at the beginning of the paper, the explanation fit better. Response to reviewers: In agreement with the reviewer’s comment and to better explain the service in the introduction section, modifications were made to include the rationale about the inclusion of the follow up. 

Modifications in the manuscript:

A sentence was moved from the comparison with the literature’s section (lines 320-321) to the introduction section (lines 79-80): MRF includes follow up due to the results found in the literature, which suggests that medication review with follow up reduces patients’ problems related with the use of medicines (18).

LINE 84: I would add an “of”. …, the assessment OF all medication….

Response to reviewers: Thank you for the correction. 

Modifications in the manuscript:

A sentence was corrected in the introduction section (line 89): […] the assessment of all medication […]

LINES 92-95. The objective of the protocol is to evaluate DRPs associated to patients´ behavior. However, the authors haven explained which type o DRPs are those. During the protocol, when talking about PharmDISC, they explain that the tool includes 24 DRPs, but only 15 are related to patients´ behavior. Again, I think this explanation should be included in the introduction.

Response to reviewers: The introduction includes the following information (lines 51-60):

Identification and management of DRPs related to health process such as drug-drug interactions, contraindications or abuse control are already remunerated as part of the "fee for services" related to prescription validation in Swiss community pharmacies (CPs) (8). Nevertheless, DRPs related to the patient and his behavior, intentional or non-intentional, (i.e., patient stores drug inappropriately, administers/uses the drug in a wrong way, etc.) need a deeper understanding of the patient’s specific situation and no service related to this aspect is currently available in Switzerland. For example, a treatment intake error could occur due to an instruction that is misunderstood / forgotten or due to improper medication storage at home. One way to identify and prevent the occurrence of DRPs in a structured manner is to conduct a medication review with the patient (9).

However, to clarify the selection of the DRPs related to patients we have included an explanation when explaining the aims of the service and a figure for the possible DRPs identified through the drug process. 

Modifications in the manuscript:

A sentence was modified in the introduction section (line 82): The aim of such service is to perform a medication review at the CP to 1) prevent, identify and manage DRPs related to patients due to the lack of such service in Switzerland, 2) […]

A figure (Figure 1) was included in the introduction section (see attached).

LINE 93: I would add an “of”. …, identification and management OF DRPs…

Response to reviewers: Thank you for the correction. 

Modifications in the manuscript:

A sentence was corrected in the introduction section (line 98): […] identification and management of DRPs associated to patients’ behavior […]

LINE 99: Work schedule should be reviewed and readjusted. (Currently we are on July, 2023)

Response to reviewers: A correction was made according to the present. 

Modifications in the manuscript:

A sentence was corrected in the study design and setting’s section (line 104): The study is taking place for 15 months between April 2023 and June 2024

LINE 111: I would change the comma to a semicolon. (…who participate in the study; for those cases, all…)

Response to reviewers: Thank you for the correction. 

Modifications in the manuscript:

A sentence was corrected in the introduction section (line 117): A CP may have different pharmacists who participate in the study; for those cases, […]

LINE 121: When talking about eligible patients, It is not clear to me the role of non-prescription drugs if only patients with 4 chronic prescription drugs (or more) can enter the study. 

Response to reviewers: No limitation on the number of non-prescription medications is included in the list of inclusion criteria.

A clarification was included in the manuscript. 

Modifications in the manuscript:

A sentence was modified in the recruitment of study participants’ section (lines 130-131): Eligible patients will be those adults taking four or more chronic drugs prescribed for at least three months prior to recruitment, irrespectively of the number of non-prescription medications.

LINE 130: I would add a “d” at the end of “include”. … the service without being included in the study.

Response to reviewers: Thank you for the correction. 

Modifications in the manuscript:

A sentence was corrected in the recruitment of study participants’ section (line 139): (patients who will not consent to participate will be able of accessing the service without being included in the study).

LINE 169. Again, the authors talk about DRPs related to patients` behavior. It should be explained in the introduction section. 

Response to reviewers: Please see reviewer’ comment and response regarding lines 92-95.

Modifications in the manuscript:

A sentence was modified in the introduction section (line 82): The aim of such service is to perform a medication review at the CP to: 1) prevent, identify and manage DRPs related to patients due to the lack of such service in Switzerland, 2) […]

A figure (Figure 1) was included in the introduction section (see attached).

LINE 180: The authors reference the PharmDISC tool using the Reference#22, but in LINE 298 they use the reference #15 to do the same.

Response to reviewers: Thank you for the correction, the reference wasn’t properly linked. It has been corrected so both sentences use the same reference (#15). 

Modifications in the manuscript:

A reference number was corrected in the data collection methods’ section (line 191): A validated tool will be used for identifying and documenting DRPs, the PharmDISC tool (15).

LINE 202: I would change the comma to a semicolon. (Cantonal Health Authority; in addition…)

Response to reviewers: Thank you for the correction. 

Modifications in the manuscript:

A sentence was corrected in the data collection methods’ section (line 213): It is planned that a short communication will be done by the Cantonal Health Authorities; in addition, […]

TABLE 1: Patients´ knowledge. The authors state that the Medication Knowledge Questionnaire is being validated, which is OK. However, I would like to know a bit more about the scoring system from -7 to 7. in this context, what would a score of zero mean? In this same box, the phrase “How do you take this medicine with food” is duplicated and should be deleted.

Response to reviewers: The score of the questionnaire is made to evaluate the evolution over time on patients’ knowledge about each medication. The lowest score is -7 where the patient has answered all questions incorrectly to +7 where he/she has answered all correctly. However, there is not limit in between to differentiate patients with appropriate knowledge. A score of 0 has no particular significance, apart from being in the middle of the two extremes. The questionnaire can be used to evaluate the same patient and his/her progress over time.

The duplicated question was deleted, thank you for the correction. 

Modifications in the manuscript:

A sentence was modified in table 1 (patient’s knowledge): Score goes from -7 (patient who answers all questions incorrectly) to 7 (patient who answer all questions correctly) for each medication that the patient is taking.

A score of 0 has no particular significance, apart from being in the middle of the two extremes. The questionnaire is being validated nowadays in a separate project. It aims to evaluate patient’s knowledge over time, rather to differentiate between patients.

A sentence was deleted in table 1 (patient’s knowledge): How do you take this medicine with food?

TABLE 1: Number of Pharmaceutical. I wonder if the term “Number of Pharmacist Interventions” should fit better than “Number of pharmaceuticals”, but as this is an English wording question, I let it to the authors to discuss about. 

Response to reviewers: “Number of pharmaceutical interventions”, “Pharmaceutical interventions classification” and “Pharmaceutical interventions acceptance” are included in the table. We could not find any variable named “Number of pharmaceuticals”.

Modifications in the manuscript: No modifications were made.

LINE 244: I would change the comma to a semicolon. …p<0.05; the software STATA… 

Response to reviewers: Thank you for the correction. 

Modifications in the manuscript:

A sentence was corrected in the statistical methods and analysis’ section (line 257): The level of significance will be set at p<0.05; the software STATA 17®

LINE 261-263: As said previously, work schedule should be reviewed and readjusted, as we currently are on the month of July.

Response to reviewers: A correction was made according to the present. 

Modifications in the manuscript:

A sentence was corrected in the status and timeline’s section (line 269): Data collection started in April 2023.

LINE 270: I would change the comma to a semicolon. … oral consent; a patient could withdrawal…

Response to reviewers: Thank you for the correction. 

Modifications in the manuscript:

A sentence was corrected in the ethics approval and safety considerations’ section (line 284): Participants will give oral consent; a patient could withdrawal […]

LINES 292-293: I would change a little bit the wording: … the National Institute for Health and Care Excellence (NICE) guidelines indicating that “all…

Response to reviewers: Thank you for the correction. 

Modifications in the manuscript:

A sentence was corrected in the discussion’s section (lines 306-307): General recommendations about the medication review service were also included in the National Institute for Health and Care Excellence (NICE) guidelines indicating that “all prescribed […]

LINE 301: I would change the comma to a semicolon. And facilitation thorough; it also includes… Response to reviewers: Thank you for the correction. 

Modifications in the manuscript:

A sentence was corrected in the discussion’s section (line 315): […] it includes pharmacists’ training at the beginning of the study and facilitation throughout; it also includes […]

LINE 301: I would change the comma to a semicolon. And facilitation thorough; it also includes… Response to reviewers: Thank you for the correction. 

Modifications in the manuscript:

A sentence was corrected in the discussion’s section (line 315): […] it includes pharmacists’ training at the beginning of the study and facilitation throughout; it also includes […]

---

## [Decision Letter · Decision Letter 1]

11 Sep 2023

Evaluation of an enhanced service for medication review with follow up in Swiss community pharmacies: pre-post study protocol

PONE-D-23-10141R1

Dear Dr. Amador-Fernández,

We’re pleased to inform you that your manuscript has been judged scientifically suitable for publication and will be formally accepted for publication once it meets all outstanding technical requirements.

Kind regards,

Sascha Köpke

Academic Editor

PLOS ONE

Additional Editor Comments (optional):

Reviewers' comments:

Reviewer's Responses to Questions

**Comments to the Author**

1. Does the manuscript provide a valid rationale for the proposed study, with clearly identified and justified research questions?

Reviewer #1: Yes

Reviewer #2: Yes

2. Is the protocol technically sound and planned in a manner that will lead to a meaningful outcome and allow testing the stated hypotheses?

Reviewer #1: Yes

Reviewer #2: Yes

3. Is the methodology feasible and described in sufficient detail to allow the work to be replicable?

Reviewer #1: Yes

Reviewer #2: Yes

4. Have the authors described where all data underlying the findings will be made available when the study is complete?

Reviewer #1: Yes

Reviewer #2: Yes

5. Is the manuscript presented in an intelligible fashion and written in standard English?

Reviewer #1: Yes

Reviewer #2: Yes

6. Review Comments to the Author

You may also provide optional suggestions and comments to authors that they might find helpful in planning their study.

Reviewer #1: All comments have been addressed.

Reviewer #2: After reviewing the authors' response to the reviewers' comments, and after re-reading and reviewing the article after the changes were made, it is as far as I am concerned ready to be published.

7. PLOS authors have the option to publish the peer review history of their article (what does this mean?). If published, this will include your full peer review and any attached files.

Reviewer #1: No

Reviewer #2: **Yes: **Miguel Angel Gastelurrutia

---

## [Editor Report · Acceptance letter]

5 Oct 2023

PONE-D-23-10141R1 

Evaluation of an enhanced service for medication review with follow up in Swiss community pharmacies: pre-post study protocol 

Dear Dr. Amador-Fernández:

I'm pleased to inform you that your manuscript has been deemed suitable for publication in PLOS ONE. Congratulations! Your manuscript is now with our production department. 

Kind regards, 

on behalf of

Professor Sascha Köpke 

Academic Editor

PLOS ONE